# Comprehensive Analysis of Exosomal MicroRNAs Derived from UVB-Irradiated Keratinocytes as Potential Melanogenesis Regulators

**DOI:** 10.3390/ijms25063095

**Published:** 2024-03-07

**Authors:** Jee-Hoe Yoon, Chan-Song Jo, Jae-Sung Hwang

**Affiliations:** Department of Genetics and Biotechnology, Graduate School of Biotechnology, College of Life Science, Kyung Hee University, Yongin 446-701, Gyeonggi-do, Republic of Korea; wlghldbs@naver.com (J.-H.Y.); jchansong93@khu.ac.kr (C.-S.J.)

**Keywords:** exosomes, microRNA, melanogenesis, melanocytes, keratinocytes, UVB

## Abstract

The exosomes derived from keratinocytes can have a substantial impact on melanogenesis by influencing melanocytes. MicroRNAs (miRNAs) encapsulated within exosomes are implicated in the control of melanogenesis, particularly when under the influence of UVB irradiation. This investigation explores UVB-induced exosomal miRNAs from keratinocytes as potential regulators of melanogenesis. UVB-irradiated, keratinocyte-derived exosomes were observed to augment melanogenesis in melanocytes, resulting in an upregulation of MITF, TRP1, TRP2, and TYR expression compared to non-UVB-irradiated exosomes. Additionally, a subset of exosomal miRNAs was differentially selected and confirmed to exert both enhancing and inhibitory effects on melanogenesis through functional assays. Notably, hsa-miR-644a, hsa-miR-365b-5p, and hsa-miR-29c-3p were found to upregulate melanogenesis, while hsa-miR-18a-5p, hsa-miR-197-5p, and hsa-miR-4281 downregulated melanogenesis. These findings suggest the involvement of keratinocyte-derived exosomal miRNAs in melanogenesis regulation within melanocytes. The expression levels of exosomal miRNAs from keratinocytes exhibited a UVB-dependent increase, indicating a potential role for these miRNAs as regulators of melanogenesis in response to UVB irradiation. Furthermore, melanogenesis was found to be dependent on exosomes derived from keratinocytes. This underscores the potential of UVB-induced exosomal miRNAs derived from keratinocytes as regulators of melanogenesis. Moreover, this study unveils a significant role for exosomes in melanocyte pigmentation, presenting a novel pathway in the intricate process of melanogenesis.

## 1. Introduction

Extracellular vesicles (EVs) are spherical, lipid-bilayered vesicles secreted by nearly all living cells [1]. They contain a variety of nucleic acids, proteins, and lipids. Exosomes, microvesicles, and apoptotic bodies are EVs found in biological fluids [2], and they have recently been shown to play a role in intra- and intercellular communication [3], as well as in numerous key physiological and pathological processes [4]. It is well established that both healthy and apoptotic cells produce EVs with diameters ranging from 30 to 10,000 nm [5,6]. Multivesicular bodies (MVBs) fuse with the plasma membrane to form exosomes, which are small extracellular vesicles (EVs) with an average size of 20–200 nm [7,8]. Small extracellular vesicles called exosomes are released by almost all cell types in both healthy and pathological states. The transmission of proteins and different nucleic acids, including microRNAs (miRNAs), mRNAs, and small regulatory RNAs (sRNAs), to nearby or distant cells can aid in cell-to-cell communication. Exosomes produced by keratinocytes have been shown to regulate melanogenesis in recent studies.

Solar radiation classified as ultraviolet B (UVB) has various documented effects on skin health. Excessive UVB exposure can lead to sunburn and skin cancer, although proper UVB exposure can modulate the immune system, for example, through systemic anti-inflammatory effects [9]. UVB light penetrates only the epidermal layer of the skin, where keratinocytes constitute over 90% of the epidermal cells. Although the underlying mechanism has not been fully explored, it is possible that mediators generated from keratinocytes are associated with the positive effects of UVB radiation. Extracellular vesicles have recently garnered increased attention due to their significance in human disorders. A newly recognized class of intercellular messengers has emerged: exosomes, a common form of an extracellular vesicle [10]. They may serve as biomarkers in dermatology due to their association with certain skin conditions [11,12]. Exosomes derived from human adipose-derived stem cells have been shown to aid in skin fibroblast proliferation and migration [13]. It has also been demonstrated that UVB irradiation of the epidermis induces the release of microvesicles both in vitro and in vivo [14,15]. Interestingly, research has revealed that keratinocyte exosomes may play a role in modulating cutaneous immunity and pigmentation [16,17]. UV irradiation triggers the release of keratinocyte exosomes to control pigmentation [18]. After exposure to UV radiation, the extracellular microvesicles produced by melanocytes stimulate keratinocytes to engage in intercellular signaling [19].

MicroRNAs (miRNAs), which are small non-coding RNAs, have been found to target mRNAs for cleavage or translational suppression, typically ranging from 19 to 24 nucleotides in length [20]. Prior to being processed and transported to the cytoplasm, miRNAs are first synthesized from intergenic regions of the genome, where they are encoded. They are subsequently broken down by the Dicer enzyme and undergo additional processing to create mature miRNAs, which are then incorporated into the RNA-induced silencing complexes involved in targeted gene silencing [19]. It is currently believed that miRNAs regulate more than 60% of protein translation through various mechanisms [21,22], highlighting the importance of miRNA research, particularly in the context of the evolving EV field. Numerous studies have demonstrated the presence of miRNAs in EVs [19,23,24]. The first study to describe this phenomenon showed that exosomes derived from mast cells contained miRNAs and demonstrated the horizontal transfer of these miRNAs between mast cells [25]. Given that miRNAs are known to regulate gene expression, it is plausible that EV miRNAs will have a similar effect in recipient cells.

Additionally, the miRNA cargo isolated from the EV populations produced by human keratinocytes has not been thoroughly studied. Moreover, the effect of exosomal miRNAs on melanogenesis in melanocytes, with or without UVB irradiation, has not been fully defined. In this study, we aimed to comprehensively analyze UVB-induced, keratinocyte-derived exosomal miRNAs as potential regulators of melanogenesis.

## 2. Results

### 2.1. NHEK Viability after UVB Irradiation

First, we investigated the cell viability of keratinocytes against UVB radiation. As a result of irradiating keratinocytes with UVB for 24 h, the cytotoxicity to UVB was significant from 100 mJ/cm^2^. Therefore, in future studies, the UVB dose for keratinocytes was determined to be 50 mJ/cm^2^ (Figure 1).

### 2.2. Isolation of Exosomes from Keratinocytes and Characterization of Exosomes

To characterize the exosomes generated from NHEKs, exosomes from keratinocyte culture supernatants were separated using ultracentrifugation. Based on TEM, exosome samples from both UVB-irradiated and non-irradiated cells included vesicles with the typical circle-shaped structure (Figure 2). A keratinocyte’s exosome is a circular membranous vesicle that ranges in size from 20 to 200 nm. The WB results showed that exosomes may express the exosomal marker proteins CD9, CD81, and calnexin. Exosomes contain the evolutionarily conserved proteins CD9 and CD81, which are popular biomarkers for exosome testing. A cell’s endoplasmic reticulum (ER) component protein, called calnexin, should not be found in exosomes (Figure 3A). The exosome size distributions, measured using an electrophoretic trace and nanoparticle tracking analysis (NTA), revealed that the above protocol’s exosome isolation yielded exosomes with a mode size of 114 ± 6.5 nm (Figure 3B,C). As a result, the particles we extracted from the keratinocyte supernatant were in fact exosomes. Based on these results, it was found that UVB irradiation did not affect the amount or size of the exosomes.

### 2.3. Effect of UVB Radiation on the Exosomal miRNA Expression of NHEKs

Next, the expression of keratinocyte-derived exosomal miRNAs was then compared before and after UVB exposure using miRNA sequencing. miRNAs whose expression levels are regulated under UVB irradiation were classified into upper and lower ranks (Figure 4A,B). Three independent experiments found 23 miRNAs with an increased expression and 42 miRNAs with a decreased expression in common in the UVB-irradiated cells compared to control (Figure 4C). The 14 common miRNAs analyzed using the program include hsa-miR-644a, hsa-miR-877-3p, hsa-miR-323a-5p, hsa-miR-125-5p, hsa-miR-138-5p, hsa-miR-579-5p, hsa-miR-365b-5p, hsa-miR-29c-3p, hsa-miR-15b-5p, hsa-miR-4281, hsa-miR-18a-5p, hsa-miR-184, hsa-miR-197, and hsa-miR-196-5p. Among the target exosomal miRNAs involved in melanogenesis, hsa-miR-644a, hsa-miR-138-5p, hsa-miR-365b-5p, hsa-miR-29c-3p, and hsa-miR-15b-5p were confirmed to increase pigmentation in melanocytes, and hsa-miR-4281, hsa-miR-18a-5p, hsa-miR-184, hsa-miR-197, and hsa-miR-196-5p were confirmed to decrease, and were screened using DIANA-microT v3.0, Target Scan, and miBase. (fold change > 1.0). These three web-based bioinformatics tools foresaw the miRNAs that would specifically target the MITF, TYR, TRP1, and TRP2. Ten of the miRNAs and inhibitors (Inh) found using the three programs were synthesized. (Figure 4C) For the predicted miRNAs list, see Table 1. As a result, all miRNAs predicted to regulate melanogenesis showed an increase in their expression compared to the control group when irradiated with UVB.

### 2.4. The Melanogenesis of Melanocytes Treated with Keratinocyte-Derived Exosomes Is Dependent on the Amount of UVB Irradiated to the Keratinocytes

We set out to determine whether they also influence the expression of specific pigmentation genes, like the master transcriptional melanogenesis regulator microphthalmia-associated transcription factor (MITF). According to the quantity of UVB radiation given to keratinocytes, we employed Western blotting to ascertain the impact of melanogenesis in melanocytes. Previously, there have been studies on the effect of melanogenesis in melanocytes according to the number of keratinocyte-derived exosomes present. However, since the effect of melanogenesis in melanocytes treated with different amounts of UVB irradiation is not yet known, the experiment was conducted. Exosomes separated by different keratinocyte UVB irradiation doses were treated with melanocytes for 48 h to confirm the expressions of MITF, TYR, TRP1, and TRP2 (Figure 5A,B). As a result, it was confirmed that melanogenesis in melanocytes significantly increased in a dependent manner according to the amount of UVB irradiation on the keratinocyte exosomes.

### 2.5. Exosomal MicroRNAs Derived from Keratinocytes Affects the Amount of Melanin in Melanocytes

We transfected melanocytes to search for the target exosomal miRNAs identified to regulate melanogenesis. Among the synthesized exosomal miRNAs, hsa-miR-644a, hsa-miR-365b-5p, and hsa-miR-29c-3p significantly increased the melanin content in melanocytes compared to the control. Exosomal miRNA inhibitors all significantly reduced the melanin contents (Figure 6A). We used 1-phenyl-2-thiourea (PTU) as a positive control in these studies because of its known inhibitory effects on melanin synthesis. Among the keratinocyte-derived exosomal miRNAs identified to downregulate melanogenesis in melanocytes, hsa-miR-18a-5p, hsa-miR-197-5p, and hsa-miR-4281 showed a significant decrease in the melanin content. This also worked significantly for all inhibitors (Figure 6B). This result suggests that keratinocyte-derived exosomal miRNAs can affect pigmentation in melanocytes.

### 2.6. The Expression Levels of Exosomal miRNAs in UVB-Irradiated Keratinocytes Is Increased

We investigated how the top three exosomal miRNAs and the bottom three exosomal miRNAs that control pigmentation in melanocytes were affected by the presence or absence of UVB irradiation in keratinocytes. Using RT-PCR, it was confirmed that the expression levels of all six exosomal miRNAs targeting melanogenesis increased significantly when UVB was irradiated into keratinocytes (hsa-miR-644a, hsa-miR-365b-5p, hsa-miR-29c-3p, hsa-miR-18a-5p, hsa-miR-197-5p, and hsa-miR-4281) (Figure 7A–D).

### 2.7. Keratinocyte-Derived Exosomal miRNAs Regulate Melanogenesis in Melanocytes

In order to look at how exosomal miRNAs affect melanogenesis in melanocytes, this process was overexpressed by the transfection of exosomal miRNA mimics and knocked down by the transfection of exosomal miRNA inhibitors. The levels of the intracellular proteins MITF, TYR, TRP1, and TRP2 were then determined. Consequently, we explored whether the primary mechanism by which exosomes controlled melanogenesis in melanocytes was based on the exosomal miRNAs derived from keratinocytes. As a result, it was confirmed that exosomal miRNAs—hsa-miR-644a, hsa-miR-365b-5p, and hsa-miR-29c-3p, which were identified to increase melanogenesis—had significantly increased protein levels compared to the control and decreased inhibitor levels (Figure 8A,B). Also, it was confirmed that exosomal miRNAs—hsa-miR-18a-5p, hsa-miR-197-5p, and hsa-miR-4281, which were confirmed to decreased melanogenesis—had significantly decreased protein expression levels to MITF, TYR, TRP1, and TRP2. The expression of the exosomal miRNA inhibitor also worked significantly (Figure 9A,B). This result suggests that keratinocyte-derived exosomal miRNAs can regulate melanogenesis in melanocytes.

## 3. Discussion

In this study, we demonstrated that UVB-irradiated, keratinocyte-derived exosomal miRNAs can control melanogenesis in melanocytes.

Melanocytes, which are dendritic cells produced from the neural crest and commonly found in the basal layer of the epidermis, are also present in the brain, heart, choroid, cochlea, and hair follicles [26,27,28]. The remarkable ability of this cell type to produce and release melanin, though well known, is often underappreciated. Through dendrites, melanocytes transmit melanin-containing melanosomes to hundreds of keratinocytes in the epidermis, forming what is known as the epidermal melanin unit [29]. Studies have shown that melanocytes communicate with Langerhans cells, endothelial cells, and peripheral neurons via their cutaneous axon terminals, contributing to various functions such as skin vascularization, sensation, and immune responses [30,31,32].

The primary physical factor affecting melanocytes, which are responsible for a significant portion of skin cancers, is UV radiation [33]. Exposure to the appropriate amount of UV light, often in the form of UVB radiation (290–320 nm), can promote vitamin D synthesis and mineral metabolism. However, excessive and prolonged UV radiation exposure can lead to inflammation, apoptosis, immune cell activation, skin cancer, and the production of various cytokines and chemokines [33,34,35]. Cells respond to UVB radiation-induced damage by altering the expressions of numerous genes involved in stress response, DNA synthesis, cell cycle regulation, and DNA repair [35,36]. UVB radiation can trigger melanocytes to produce hormones such as pro-opiomelanocortin (POMC), melanocyte-stimulating hormone (a-MSH), basic fibroblast growth factor (bFGF), and adrenocorticotrophic hormone to regulate melanogenesis and other stress responses. Melanocytes act as epithelial “stress sensors”, protecting not only themselves but also other skin cells from UV damage [37]. While melanogenesis serves as the primary defense mechanism of melanocytes against environmental threats, it is undoubtedly not the sole or most important mechanism.

Exosomes, which are small intracellular vesicles with diameters of 20–200 nm, have been shown in several studies to control melanocyte melanogenesis when generated by keratinocytes. However, it remains unclear whether the exosomal miRNAs released by skin keratinocytes play a role in controlling melanogenesis. Upon contact with melanocytes, exosomes and extracellular vesicles (EVs) from keratinocytes increase TYR activity and the expression of pigmentation genes, resulting in an increased melanin concentration in the recipient melanocytes. Our results also indicate that UVB radiation induces keratinocytes of various phototypes to release exosomes, which regulate melanocyte pigmentation to varying degrees, likely through different processes.

Keratinocytes can release extracellular vesicles (EVs), and investigations into the components of exosomes derived from melanoma cells have utilized the miRNAs found in the exosomes of normal human epidermal keratinocytes as a reference. We focused on the potential of keratinocytes to release EVs that could impact other cells and successfully isolated the exosomes produced by keratinocytes. It is crucial to consider the technical processes used to separate, purify, and quantify the exosomes released by keratinocytes for data standardization. Our findings furthered our understanding of the properties of the EVs released by keratinocytes; we used keratinocyte cell-culture supernatants to generate vesicles expressing proteins found in the intraluminal vesicles (ILVs) of multivesicular bodies (CD81, CD9, and calnexin).

In this study, we discovered that UVB-irradiated miRNAs may enhance the secretory function of melanocytes and alter their exosomal miRNA profile, indicating the cellular response to irradiation and promoting melanogenesis in melanocytes. A significant portion of the miRNAs circulating in the body are present in exosomes and are involved in numerous intercellular communication pathways [36]. Thus, we investigated the changes in melanocyte exosomal miRNA profiles before and after UVB exposure and observed the distinct patterns of the exosomal miRNA response to UVB exposure. We examined how keratinocyte-derived exosomal miRNA expression changed after UVB exposure and identified specific target genes of exosomal miRNAs in response to UVB exposure. Among the top three expressed miRNAs in exosomes derived from UVB-irradiated keratinocytes, three miRNAs upregulated melanogenesis (hsa-miR-644a, hsa-miR-365b-5p, and hsa-miR-29c-3p) while three exosomal miRNAs downregulated melanogenesis (hsa-miR-18a-5p, hsa-miR-197-5p, and hsa-miR-4281).

These findings suggest that keratinocyte-derived exosomal miRNAs are involved in melanogenesis in melanocytes. The expression levels of exosomal miRNAs from keratinocytes increase depending on the level of UVB irradiation, and melanogenesis is dependent on the exosomes derived from keratinocytes. This raises the possibility of exosomal miR-NAs derived from UVB-irradiated keratinocytes being potential regulators of melanogenesis. Furthermore, this study revealed an important role of exosomes in melanocyte pigmentation, opening up a new pathway in melanogenesis. It is interesting that some miRNAs derived from UVB-irradiated keratinocytes suppressed melanogenesis. However, considering that since among the factors induced in keratinocytes by ultraviolet radiation, certain factors such as alpha-MSH and endothelin-1, for example, stimulate melanin formation while others such as interleukin-6 (IL-6) and transforming growth factor (TGF)-β, for example, decrease it, it seems plausible that this could be possible [38].

Although this study found that specific miRNAs are involved in melanogenesis, the exact mechanism by which these miRNAs exert their effects is still unclear. Additional research is needed to elucidate these mechanisms by examining how they interact with the melanogenic gene pathway. Additionally, although our studies have identified a variety of miRNAs with this regulatory potential, we recognize that the lack of in vivo data limits our ability to conclusively determine their biological significance. To address these gaps, future studies need to utilize advanced experimental approaches, such as 3D epidermal organotypic cultures and human biopsies of UVB-irradiated skin with protected skin controls. This approach would allow us to examine the differences in the exosome controls, miRNA content, and melanin expression in a more physiologically relevant context. These methods could validate our findings and represent an important step toward uncovering the true biological and therapeutic implications of exosomal miRNAs in the regulation of melanogenesis.

## 4. Materials and Methods

### 4.1. Cell Culture

Normal human epidermal melanocytes (NHEMs) were cultured in Medium 254 (M-254-500; Invitrogen Life Technologies, Carlsbad, CA, USA) with 1% penicillin/streptomycin and human melanocyte growth supplement (HMGS; Invitrogen Life Technologies, Carlsbad, CA, USA) at 37 °C in a humidified atmosphere of 5% CO_2_.

Neonatal-derived normal human epidermal keratinocytes (NHEKs) were purchased from Invitrogen (Carlsbad, CA, USA). NHEKs were grown in EpiLife^®^ media (Life Technologies, Carlsbad, NY, USA) containing human keratinocyte growth supplement, 1% penicillin/streptomycin, and 60 μM CaCl_2_ (Invitrogen). Cells were kept in an incubator with 5% CO_2_ at 37 °C.

### 4.2. Exosome Isolation and Preparation

Normal human keratinocytes (NHEKs) were maintained in an exosome-free medium after UVB irradiation of 50mJ/cm^2^. Exosomes were extracted using ultracentrifugation from the keratinocyte culture medium after two days of incubation. To quickly eliminate impurities, the material was filtered using a 0.22 μm syringe filter. After that, the debris was removed by centrifuging the medium at 1000 rpm for 10 min at 4 °C. The debris was then removed once more by centrifuging the supernatant for 90 min at 4 °C at 10,000 rpm. Additionally, the supernatant was centrifuged twice for 120 min (4 °C) at 25,000 rpm to separate the exosomes from it. The pelleted exosomes were re-suspended in PBS, which has a pH range of 7.2 to 7.4, and stored at −80 °C.

### 4.3. MicroRNAs (miRNAs)

Exosomal miRNAs were purchased from Bioneer (Daejeon, Republic of Korea), and the potential melanogenesis target genes of the miRNAs were predicted using DIANA-microT v4.0 (https://dianalab.e-ce.uth.gr/html/universe/index.php?r=microtv4, accessed on 3 May 2023) miRBase (https://www.mirbase.org/, accessed on 5 May 2023), and TargetScan (https://www.targetscan.org/vert_80/, accessed on 3 May 2023).

hsa-miR-644a mimics 5′-AGUGUGGCUUUCUUAGAGC-3′;

hsa-miR-365b-5p mimics 5′-AGGGACUUUCAGGGGCAGCUGU-3′;

hsa-miR-29c-3p mimics 5′-UGACCGAUUUCUCCUGGUGUUC-3′;

hsa-miR-18a-5p mimics 5′-UAAGGUGCAUCUAGUGCAGAUAG-3′;

hsa-miR-197-5p mimics 5′-CGGGUAGAGAGGGCAGAGGGAGG-3′;

has-miR-4281 mimics 5′-GGGUCCCGGGGAGGGGGG-3′;

hsa-miR-196a-1 mimics 5′-UAGGUAGUUUCAUGUUGUUGGG-3′;

hsa-miR-15b-5p mimics 5′-UAGCAGCACAUCAUGGUUUACA-3′;

hsa-miR-138-5p mimics 5′-AGCUGGUGUUGUGAAUCAGGCCG-3′;

hsa-miR-184 mimics 5′-UGGACGGAGAACUGAUAAGGGU-3′.

### 4.4. MicroRNA Transfection

NHEMs were seeded in 6-well plates 24 h prior to transfection. The media were replaced with antibiotic-free media 1 h before transfection. Synthetic miRNA mimics (Bioneer, Daejeon, Republic of Korea) were transfected into cells using the Lipofectamine 2000 purchased from Invitrogen (Carlsbad, CA, USA); the reagent was applied according to the manufacturer’s protocol. After 8 h, the miRNA inhibitor was transfected into the cells using the same protocol. And after 24 h, the media were replaced with Medium254 that contained human melanocyte growth supplement and 1% penicillin/streptomycin.

### 4.5. Cell Viability Analysis

NHEKs were seeded in a 96-well plate and stabilized at 37 °C for 24 h. Each well was then rinsed with Dulbecco’s Phosphate Buffered Saline (DPBS), and LPS was added. After 72 h, the cells were washed with DPBS and replenished with a medium containing 10% EZ-Cytox (Daeil Laboratories, Yongin, Republic of Korea). Using a microplate reader, the absorbance of each well was measured at 450 nm following an additional incubation at 37 °C for 30 min to 1 h (TECAN, Zürich, Switzerland).

### 4.6. Ultraviolet B Treatments

Only normal human epidermal keratinocytes from the second to fourth passage were used. Keratinocytes were seeded and irradiated 24 h later with one shot of 50 mJ/cm^2^ of ultraviolet B (312 nm) using a Biosun machine (Vilber Lourmat, Suarlée, Belgium). The culture medium was washed right away and replaced with PBS before being exposed to radiation.

### 4.7. Melanin Contents Assay

Normal human epidermal melanocytes were seeded in a 24-well plate, and the next day, the cells were washed in DPBS and transfected with the hsa-miR-644a mimic, hsa-miR-365b-5p mimic, hsa-miR-29c-3p mimic, hsa-miR-644a inhibitor, hsa-miR-365b-5p inhibitor, hsa-miR-29c-3p inhibitor, hsa-miR-18a-5p mimic, hsa-miR-197-5p mimic, hsa-miR-4281 mimic, hsa-miR-18a-5p inhibitor, hsa-miR-197-5p inhibitor, and hsa-miR-4281inhibitor in Medium 254 that contained human melanocyte growth supplement for 48 h. After washing with DPBS, cells were lysed in 1N NaOH for 20 min on a plate shaker. The resulting cell lysate was transferred to a 96-well plate for an optical density (OD) measurement at 490 nm using a microplate reader. Keraskin-M was dissolved in 1N NaOH and sonicated for the purpose of melanin analysis using 3D human tissue reconstruction. By centrifuging the debris at 13,000 rpm for 5 min, the debris was cleared, and the supernatant’s OD was then measured at 490 nm using a microplate reader. The protein concentration of the cell lysates was determined using the BCA Protein Assay Reagent Kit (Pierce Biotechnology, Inc., Rockland, IL, USA) standardized with bovine serum albumin.

### 4.8. Reverse Transcription-Polymerase Chain Reaction (RT-PCR)

Cells were lysed using trizol reagent (Takara, Otsu, Japan) in normal human epidermal keratinocytes, and chloroform was added as much as 1/5 of the amount of trizol reagent, and centrifugation was performed at 4 °C at 13,000 rpm for 15 min. After that, only the supernatant was transferred, and the same amount of isopropylethanol was added. After that, centrifugation was performed for 20 min under the same conditions, and only the pellet was removed. After adding and removing DEPC-EtOH, the supernatant was finally diluted with DEPC-water. The purity and amount of total RNA was determined using a NanoDrop2000 (Thermo Scientific: Waltham, MA, USA). For the CDNA synthesis, 2 ug of total RNA was mixed with oligo (dT) or random hexamers (ELPIS: Daejeon, Republic of Korea), denatured at 65 °C for 5 min, and cooled on ice for 5 min. The reverse transcriptase and 2 mM dNTP (Fermentas: Waltham, MA, USA) were then added to the annealed samples, and they were incubated at 42 °C for 1 h. The reverse transcription was terminated by heating at 70 °C for 10 min. For PCR, the HiPi PCR Mix (ELPIS) was used for the remaining materials except for the cDNA and primer.

hsa-miR-644a sense 5′-AGUGUGGCUUUCUUAGAGC-3′ andantisense 5′-GCUCUAAGAAAGCCACACU-3′.hsa-miR-365b-5p sense 5′-AGGGACUUUCAGGGGCAGCUGU-3′ andantisense 5′-ACAGCUGCCCCUGAAAGUCCCU-3′.hsa-miR-29c-3p sense 5′-UGACCGAUUUCUCCUGGUGUUC-3′ andantisense 5′-GAACACCAGGAGAAAUCGGUCA-3’.hsa-miR-18a-5p sense 5′-UAAGGUGCAUCUAGUGCAGAUAG-3′ andantisense 5′-CUAUCUGCACUAGAUGCACCUUA-3′.hsa-miR-197-5p sense 5′-CGGGUAGAGAGGGCAGUGGGAGG-3′ andantisense 5′-CCUCCCACUGCCCUCUCUACCCG-3′.hsa-miR-4281 sense 5′-GGGUCCCGGGGAGGGGGG-3′ andantisense 5′-CCCCCCUCCCCGGGACCC-3′.

An electrophoretic separation on 2% agarose gels and staining with RedSafeTM Nucleic Acid Staining Solution were used to visualize the PCR products that were produced (ELPIS).

### 4.9. Transmission Electron Microscopy (TEM)

Ten μL of the exosome solution was dropped onto a copper mesh for an electron microscopy examination. After using filter paper to drain the extra liquid, a staining solution of 2% phosphotungstic acid was applied. Under the transmission electron microscope, the morphology was examined (Hitachi, Tokyo, Japan)

### 4.10. Nanoparticle Tracking Analysis (NTA)

According to what we previously reported, the NS300 (Nanosight: Amesbury, UK) was used to determine the sizes and concentrations of the isolated EXs. In a nutshell, sterile-filtered PBS was used to dilute the EX samples to a concentration of 10^7^–10^8^ particles mL^−1^. Following the dilution, 700 mL of the sample was put into the device for movement tracking at a rate of 30 frames s^−1^. The videos of the moving particles were captured at least three times for each sample at different angles, and the NTA software then examined them (version 2.3, Nanosight). The precise dilution factor for the NTA results was taken into account before determining the particle concentration. The findings of the NTA analysis were calculated as the average of the three tests that were run for each sample.

### 4.11. Western Blot Analysis

Normal human epidermal melanocytes were prepared with RIPA solution (Noble Bio, Hwaseong-si, Republic of Korea), protease inhibitor cocktail (Sigma, St. Louis, MO, USA), and 1 mM phenylmethanesulfonyl fluoride (PMSF: Sigma, St. Louis, MO, USA) at 37 °C for 30 min. The solutes dissolved in minutes. After centrifuging the lysate at 13,000 rpm for 20 min, the supernatant was separated. The protein content of the supernatant was subjected to a BCA analysis using the Pierce Protein Assay Kit (Thermo Scientific, Rockford, IL, USA). After SDS-PAGE, blotting was performed on the PVDF membrane and washed 3 times with TBS-T buffer. After blocking in TBS-T mixed with 5% BSA for 1 h, the membrane was treated with primary antibody overnight. The antibodies used were as follows: B-actin antibody (Sigma, St. Louis, MO, USA), MITF antibody (ThermoFisher Scientific, Fremont, CA, USA), rabbit anti-tyrosinase (Abcam, Cambridge, UK: ab170905), rabbit anti-TRP1 (Abcam, ab178676), rabbit anti-TRP2/DCT (Abcam, ab221144), anti-CD63 antibody (mouse monoclonal to CD63, ab59479), anti-human CD9 IgG rabbit polyclonal (System Biosciences, Palo Alto, CA, USA), and anti-human calnexin rabbit polyclonal (Cell signaling Technology, Danvers, MA, USA, Beverly; National Institute of Health, Bethesda, MD, USA). The next day, the blots were washed three times with TBS-T and reacted with horseradish peroxidase-conjugated anti-rabbit (Bethyl Laboratories, Montgomery, AL, USA) or anti-mouse antibody (BioRad, Hercules, CA, USA) for 2 h at room temperature. The bound antibodies were detected using the WEST-ZOL^®^ Plus Western Blot Detection System (INtRON Biotechnology, Seongnam, Republic of Korea) and SuperSignalTM West Pico PLUS Chemiluminescent Substrate (Thermo Scientific, Rockford, IL, USA). The bands on the membrane were detected using chemiluminescence and visualized with the Chemi Doc XRS (Bio-Rad, Hercules, CA, USA) and FluorChem E (Protein Simple, San Jose, CA, USA).

## 5. Conclusions

In conclusion, in this study, we demonstrated that UVB-irradiated, keratinocyte-derived exosomal miRNAs can control melanogenesis in melanocytes. And the melanogenesis is dependent on the exosomes derived from keratinocytes. This leads to the possibility of exosomal miRNAs derived from UVB-irradiated keratinocytes being potential melanogenesis regulators. Furthermore, this study revealed an important role for exosomes in melanocytes pigmentation, which opens a new pathway in melanogenesis.

## Figures and Tables

**Figure 1 ijms-25-03095-f001:**
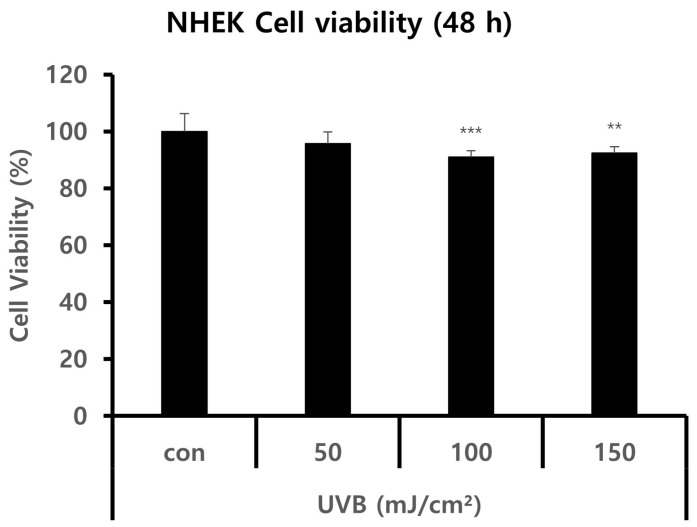
Cell viabilities of NHEKs for different UVB irradiation amounts. The cell viability was detected using an EZ-Cytox assay. The NHEK cultures were treated with 50 mJ/cm^2^, 100 mJ/cm^2^, and 150 mJ/cm^2^ UVB radiation for 48 h. The cell was then incubated with EZ-Cytox reagent, after which the cell viability was assessed. There was no change in the cell viability when irradiated with UVB at 50 mJ/cm^2^. The results are displayed as the mean ± SD. The significant differences from the control cell are labeled with asterisk, ** *p* < 0.01, and *** *p* < 0.001.

**Figure 2 ijms-25-03095-f002:**
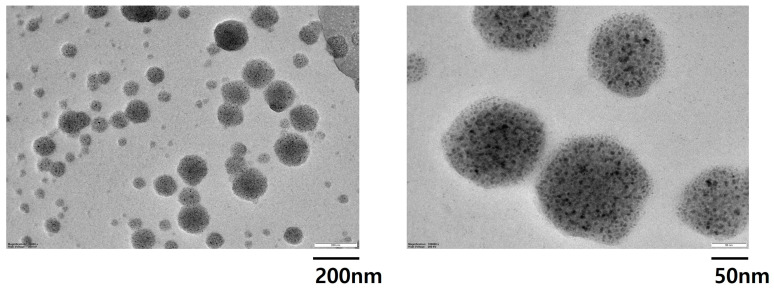
Morphology of exosomes derived from keratinocytes. Transmission electron microscopy (TEM) images display the morphology and size of the exosomes, which are negatively stained. The scale bar represents 200 nm and 50 nm.

**Figure 3 ijms-25-03095-f003:**
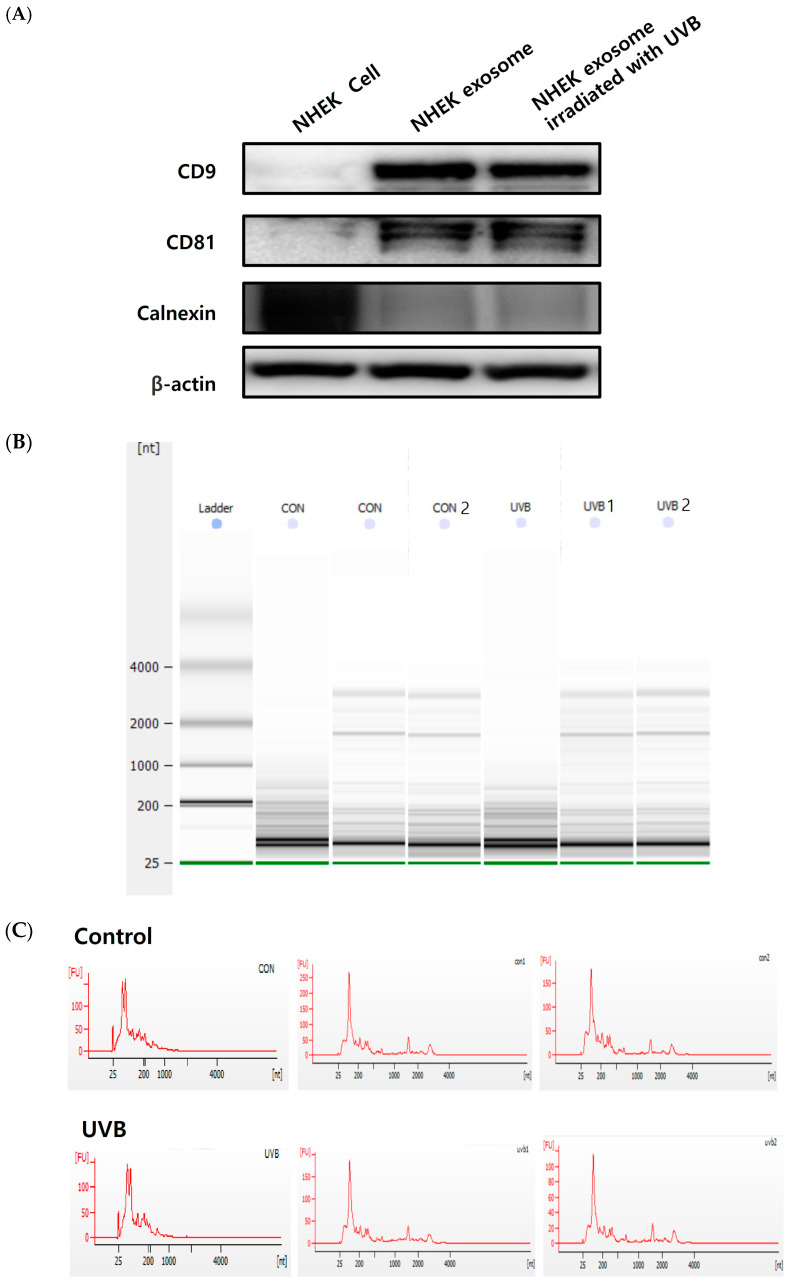
Exosomes derived from UVB-irradiated keratinocytes are isolated and characterized by their size. (**A**) Confirmation of exosome specific markers such as CD9 and CD81 using Western blot. Negative exosomal marker calnexin was lost in the samples. (**B**) Exosome size distribution measured using electrophoretic trace. (**C**) Size distribution of microvesicles (nm) assessed using nanoparticle tracking analysis. The NTA results of keratinocyte cell-derived exosomes showed a 114 ± 6.5 nm mode size of the exosomes.

**Figure 4 ijms-25-03095-f004:**
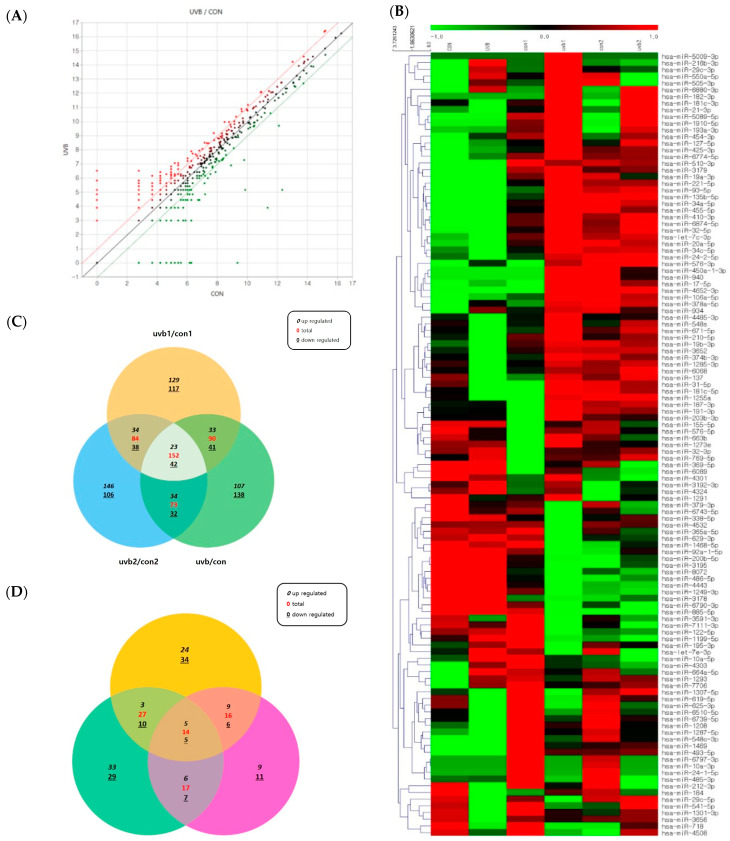
The expression levels of the identified microRNAs (miRNAs) in UVB-irradiated, keratinocyte-derived exosomes (UVB group) and non-irradiated, keratinocyte-derived exosomes (Con group). (**A**) Network analysis of regulated miRNAs and their target genes. Red dots represent the upregulated miRNAs and green dots represent the downregulated miRNAs. (**B**) Heat map representation of small RNA sequencing data about the expression levels of identified miRNAs in the UVB and Con groups. The data are from three independent experiments in each group. The melanocytes in the UVB group were treated with 50 mJ/cm^2^ of UVB radiation. The upregulated miRNAs are depicted in shades of red; downregulated miRNAs are in shades of green (fold change > 1.0). (**C**) Venn diagrams of data from each independent experiment. Expression levels of common miRNAs in the control group and the UVB irradiation group. (**D**) Number of target miRNAs that were predicted to affect melanogesis, predicted using DIANA-microT v4.0, Target Scan, and miBase. Yellow area represents DIANA-microT v3.0 database, green represents TargetScan, and pink represents miBase. The three databases together predict the upregulation of melanogenesis by 5 miRNAs, and downregulation of melanogenesis by 5 miRNAs (at the center of the diagram).

**Figure 5 ijms-25-03095-f005:**
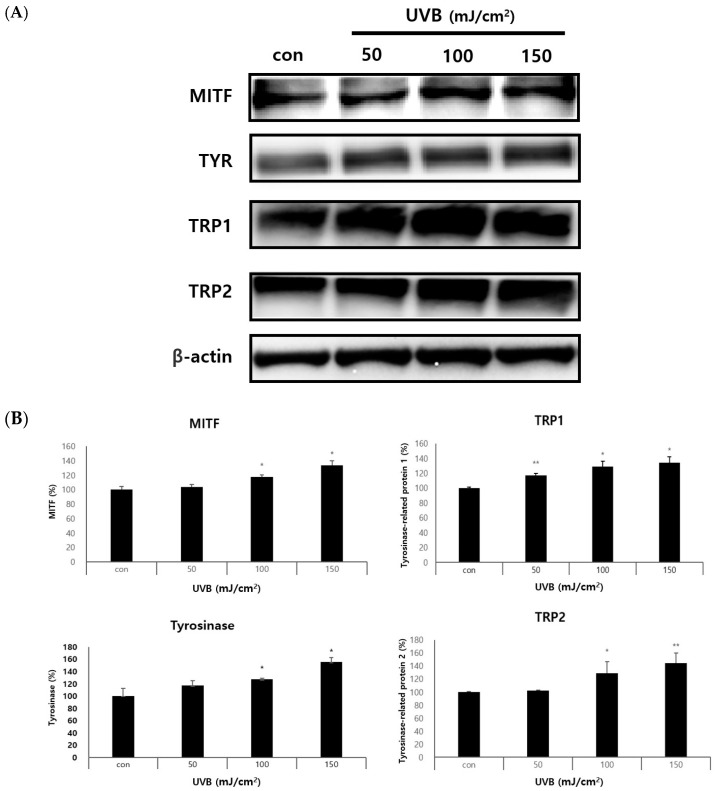
Expressions of melanogenesis target genes of keratinocyte-derived exosomal miRNAs according to UVB irradiation dose. (**A**,**B**) Melanocytes were treated with UV-irradiated, keratinocyte-derived exosomes and cultured for 72 h. Western blots was used to analyze the expressions of MITF, TYR, TRP1, and TRP2 in melanocytes. MITF, TYR, TRP1, and TRP2 all significantly increased their protein expression as keratinocyte-derived exosomes were treated with high UVB irradiation. The results are displayed as the mean ± SD. The significant differences from the control cell are labeled with asterisk, * *p* < 0.05, and ** *p* < 0.01.

**Figure 6 ijms-25-03095-f006:**
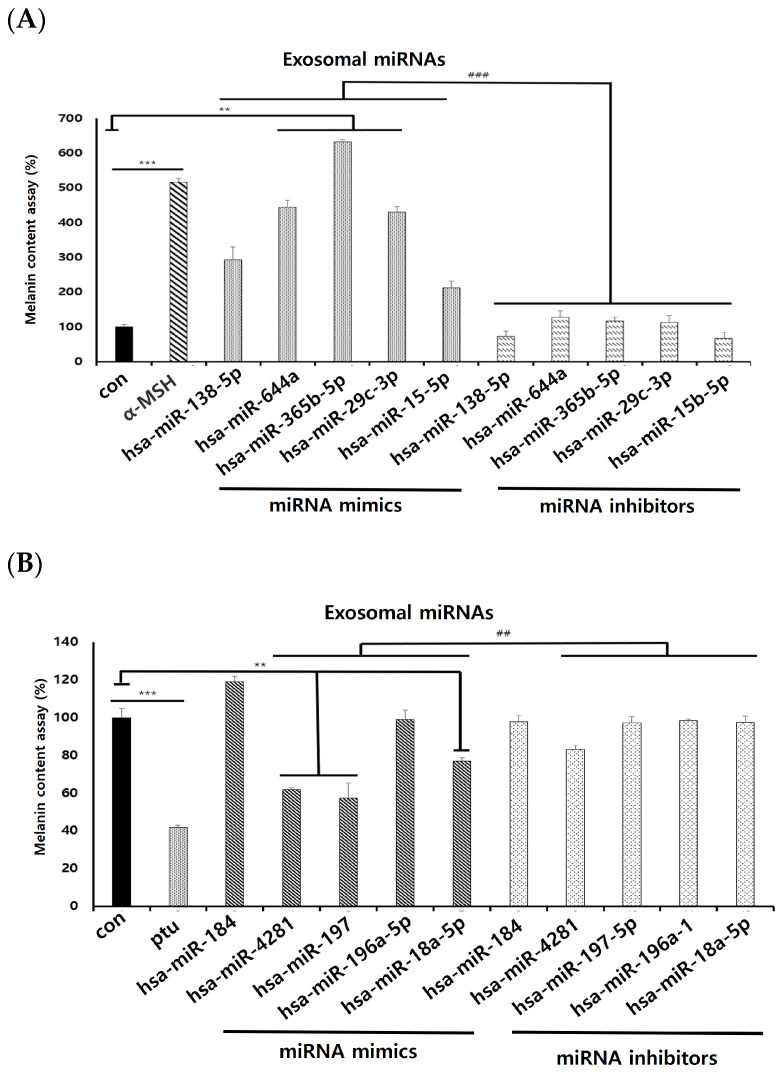
Melanin contents of melanocytes transfected with keratinocyte-derived exosomal miRNAs. Exosomal miRNAs can be expressed in melanocytes. Exosomal miRNAs derived from keratinocytes were incubated with melanocytes. Melanin contents of cells transfected with exosomal miRNAs were determined. Keratinocyte-derived exosomal miRNAs were transfected into melanocytes. (**A**) Identification of exosomal miRNAs that increases melanogenesis. (**B**) Identification of exosomal miRNAs that reduces melanogenesis. The experiment was repeated three times. The results are displayed as the mean ± SD. The significant differences from the control cell are labeled with asterisk, ** *p* < 0.01, and *** *p* < 0.001. The significant differences from the miRNA inhibitors are labeled with hashtag, ## *p* < 0.01, and ### *p* < 0.001.

**Figure 7 ijms-25-03095-f007:**
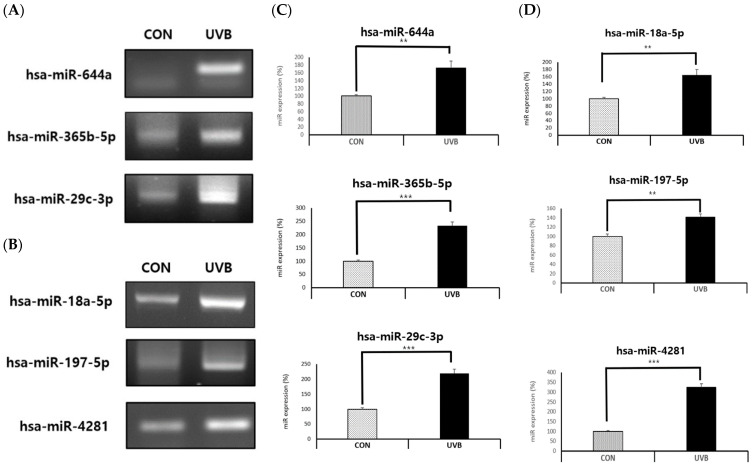
Expressions of exosomal miRNAs in keratinocytes with and without UVB irradiation. RT-PCR analysis of exosomal miRNA expression in NHEKs. NHEKs were seeded at 3 × 10^5^ cells/well in 10π dishes with EPI-life medium. After 24 h, NHEKs were irradiated with UVB at 50 mJ/cm^2^. Then, the cells were treated with exosomal miRNAs. After transfection for 72 h, expression levels were determined using RT-PCR. The experiment was repeated three times. (**A**,**C**) Target miRNAs identified to increase melanogenesis. (**B**,**D**) Target miRNAs identified to reduce melanogenesis. The results are displayed as the mean ± SD. The significant differences from the control cell are labeled with asterisk, ** *p* < 0.01, and *** *p* < 0.001.

**Figure 8 ijms-25-03095-f008:**
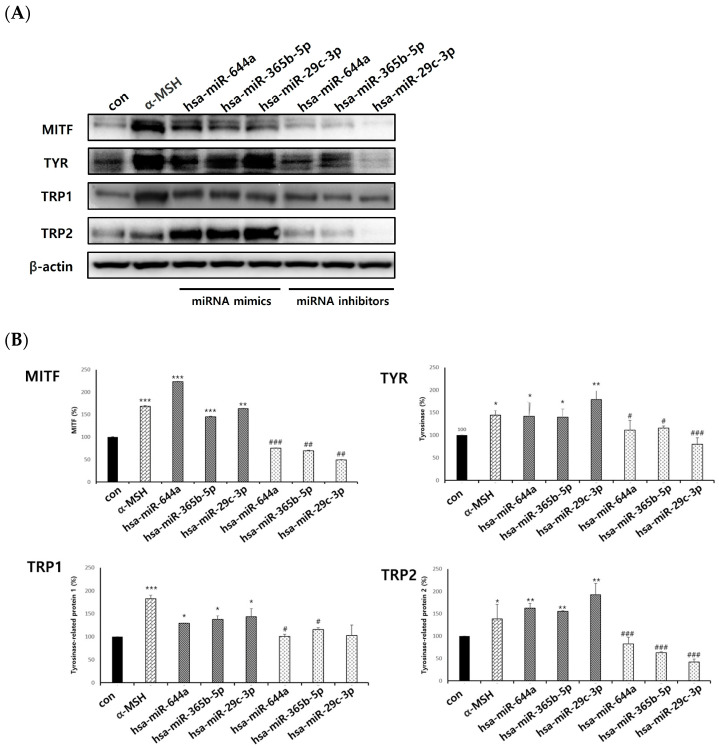
Protein levels of exosomal miRNAs with increased expressions of melanogenesis genes. Exosomal miRNAs that increase melanogenesis genes are shown. (**A**,**B**) NHEMs were transfected with hsa-miR-644a mimic, hsa-miR-365b-5p mimic, hsa-miR-29c-3p mimic, hsa-miR-644a inhibitor, hsa-miR-365b-5p inhibitor, and hsa-miR-29c-3p inhibitor, respectively. Expressions of MITF, TYR, TRP1, and TRP2 were measured using a Western blot. The experiment was repeated three times. The results are displayed as the mean ± SD. The significant differences from the control cell are labeled with asterisk, * *p* < 0.05, ** *p* < 0.01, and *** *p* < 0.001. The significant differences between the miRNA mimics and miRNA inhibitors are labeled with hashtag, # *p* < 0.05, ## *p* < 0.01, and ### *p* < 0.001.

**Figure 9 ijms-25-03095-f009:**
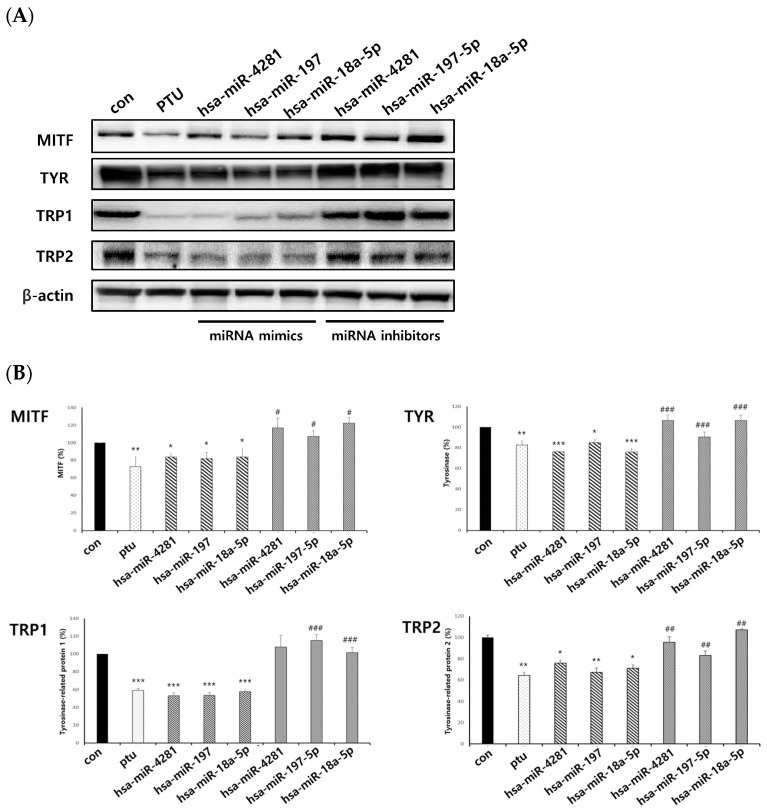
Protein levels of exosomal miRNAs with decreased expressions of melanogenesis genes. Exosomal miRNAs that decrease melanogenesis genes are shown. (**A**,**B**) NHEMs were transfected with hsa-miR-18a-5p mimic, hsa-miR-197-5p mimic, hsa-miR-4281 mimic, hsa-miR-18a-5p inhibitor, hsa-miR-197-5p inhibitor, and hsa-miR-4281inhibitor, respectively. Expressions of MITF, TYR, TRP1, and TRP2 were measured using a Western blot. In this study, 1-phenyl-2-thiourea (PTU), which is known to have an inhibitory effect on melanin synthesis, was used as a positive control. The experiment was repeated three times. The results are displayed as the mean ± SD. The significant differences from the control cell are labeled with asterisk, * *p* < 0.05, ** *p* < 0.01, and *** *p* < 0.001. The significant differences between the miRNA mimics and miRNA inhibitors are labeled with hashtag, # *p* < 0.05, ## *p* < 0.01, and ### *p* < 0.001.

**Table 1 ijms-25-03095-t001:** MicroRNAs predicted to regulate melanogenesis.

Upregulate	Downregulate
hsa-miR-644a	hsa-miR-18a-5p
hsa-miR-365b-5p	hsa-miR-197-5p
hsa-miR-29c-3p	hsa-miR-184
hsa-miR-138-5p	hsa-miR-4281
hsa-miR-15-5p	hsa-miR-196a-5p

## Data Availability

The data presented in this study are available on request from the corresponding author.

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
