# Peer review of "Comprehensive Analysis of Exosomal MicroRNAs Derived from UVB-Irradiated Keratinocytes as Potential Melanogenesis Regulators"

_ijms, 2024, doi:10.3390/ijms25063095_

Round 1
Reviewer 1 Report
Comments and Suggestions for Authors
This manuscript presents a detailed study on the role of exosomal microRNAs (miRNAs) derived from keratinocytes under UVB irradiation and their potential regulatory effects on melanogenesis in melanocytes.
To enhance the manuscript, consider the following suggestions for revision and improvement.
1. Clarification of Mechanisms: While the study identifies specific miRNAs that regulate melanogenesis, the exact mechanisms through which these miRNAs exert their effects remain unclear. Expanding on how these miRNAs interact with melanogenic genes and pathways would strengthen the findings.
2. Comparative Analysis with Other Studies: Integrating findings from this study with existing literature on miRNA regulation of melanogenesis could provide a more comprehensive view of the field. Discussing similarities and differences with previous studies might help in positioning this study within the broader context of melanocyte biology and pigmentation research.
Reviewer 2 Report
Comments and Suggestions for Authors
Major points
1. In general, UVB irradiation is known to increase melanogenesis. What is the significance of the expression of miRNAs that downregulate melanogenesis? Could you discuss that?
2. Check and remake Graphs presented in Figure 8 and Figure 9. Minor points
Minor points
1. Subtitle of Result 2.4, “The melanogenesis of melanocytes treated with keratinocyte-derived exosomes is expressed dependently on the amount of UVB irradiated to the exosome” is confused. “UVB irradiated to the exosome” should be changed to “ UVB irradiated to the keratinocytes”
2. In most of the graphs used in this manuscript, the unit of the axis is omitted, or the resolution or font size of the graph is small, making it difficult to understand information about the axis.
3. a-MSH --> α-MSH in Figure 6 and Figure 8
4. b-actin --> β-actinin all Western blot data
8. Describe “ptu” in Figure 6 and Figure 9 in each legend.
Comments on the Quality of English Language
This manuscript needs to be reviewed overall for typos and grammar.
Reviewer 3 Report
Comments and Suggestions for Authors
There are only 37 references, not 38. The alleged reference 20 is just the journal info for reference 19. Please correct.
On page 12, lane 333 the authors say that they used 60 M CaCl2 in the NHEK medium. That is as obvious typo. What was the actual CaCl2 concentration used? Was that concentration suggested by the company (Invitrogen) or selected by the authors?
The conclusion and the abstract needs to explicitly state that various miRs derived from UVB-irradiated keratinocytes can regulate melanin expression. No data are presented regarding the actual contribution or significance of this regulatory potential in vivo. An initial step towards this could be using 3D epidermal organotypic cultures as source of the exosomes. In addition, human biopsies from UVB irradiated skin with protected skin controls could be examined for exosome control, miR content and melanin expression differences to prove the significance of this regulatory potential in vivo. These would be expensive and time-consuming approaches, so in the absence of those the authors should rephrase their conclusions in a less categorical language regarding the actual biological or treatment role they conclude.
Comments on the Quality of English LanguageModerate proofreading would be helpful.
